# *Drosophila*: An Important Model for Exploring the Pathways of Inflammatory Bowel Disease (IBD) in the Intestinal Tract

**DOI:** 10.3390/ijms252312742

**Published:** 2024-11-27

**Authors:** Chuisheng Zeng, Fengying Liu, Yuhan Huang, Qianqian Liang, Xiaohong He, Lingzhi Li, Yongfang Xie

**Affiliations:** 1Institute of Bioinformatics, Chongqing University of Posts and Telecommunications, Chongqing 400065, China; zengcs@cqupt.edu.cn (C.Z.); liu.feng.ying@outlook.com (F.L.); hyuhan04@outlook.com (Y.H.); liangdoubleq@outlook.com (Q.L.); hexh@cqupt.edu.cn (X.H.); lilz@cqupt.edu.cn (L.L.); 2Jinfeng Laboratory, Chongqing 400065, China

**Keywords:** inflammatory bowel disease, *Drosophila*, signaling pathways, natural products

## Abstract

Inflammatory bowel disease (IBD) is a chronic and recurring lifelong condition, the exact etiology of which remains obscure. However, an increasing corpus of research underscores the pivotal role of cellular signaling pathways in both the instigation and management of intestinal inflammation. *Drosophila*, owing to its prodigious offspring, abbreviated life cycle, and the conservation of signaling pathways with mammals, among other advantages, has become a model organism for IBD research. This review will expound on the feasibility of utilizing *Drosophila* as an IBD model, comparing its intestinal architecture with that of mammals, its inflammatory responses, and signaling pathways. Furthermore, it will deliberate on the role of natural products across various biological models of IBD pathways, elucidating the viability of fruit flies as IBD models and the modus operandi of cellular signaling pathways in the context of IBD.

## 1. Introduction

Inflammatory bowel disease (IBD) is a chronic inflammatory condition mediated by immune responses [1], which includes two principal forms: Crohn’s disease (CD) and ulcerative colitis (UC). These disorders share similar clinical manifestations, such as abdominal pain, diarrhea, and rectal bleeding. In the context of noxious stimuli and microbial incursion, inflammation represents a complex protective mechanism against biological and chemical insults; although advantageous, sustained inflammation can induce cellular damage, leading to a variety of diseases, including IBD and colorectal cancer (CRC) [2].

The etiology of IBD involves a complex interplay of genetic predisposition, disruption of intestinal homeostasis, dysregulation of the microbiome, and heightened oxidative stress [3]. Although the precise pathogenesis of IBD remains to be fully elucidated, burgeoning research suggests that the modulation and perturbation of intestinal homeostasis are intricately linked to signaling pathways associated with IBD, such as the Janus kinase/signal transducer and activator of transcription (JAK/STAT), wingless/Wnt (Wnt/Wg), Toll-like receptor 4/nuclear factor kappa-B (TLR4/NF-κB), and Notch pathways [4]. Pro-inflammatory mediators, cytokines, and transcription factors associated with these pathways play pivotal roles in the pathogenesis of IBD [5]. IBD is distinguished by recurrent mucosal inflammation that inflicts damage upon the intestinal epithelium and impedes the structural and functional integrity of the barrier. The intestinal epithelial barrier relies on the self-renewal and proliferation of intestinal stem cells (ISCs), a process that is governed by the modulation of various signaling pathways [6].

In IBD, there is a pronounced augmentation of intestinal oxidative stress, concurrent with a diminution in the antioxidant capacity of the endogenous antioxidant system, resulting in oxidative stress-induced damage to macromolecules. Investigations have revealed elevated levels of reactive oxygen species (ROS) within the inflamed intestinal segments of IBD patients, alongside a substantial aggregation of leukocytes at these inflammatory foci, generating pro-inflammatory cytokines and excessive ROS [7]. This contributes to heightened intestinal oxidative stress, which subsequently leads to impairment of the mucosal barrier’s structural and functional integrity. The manifestation of pro-inflammatory cytokines and ROS is mediated by associated signaling pathways. This underscores the significance of signaling pathways in IBD and suggests that their investigation could elucidate the pathogenic mechanisms underlying IBD.

*Drosophila melanogaster* is a well-established model organism for investigating human intestinal disorders, boasting the advantages of high conservation and cost-effectiveness. There is a significant conservation of basic physiological and functional characteristics between *Drosophila* and humans [8], with approximately 75% homology in disease-associated genes [4]. The intestinal architecture and functionality of *Drosophila* exhibit parallels to those of mammals, and the signaling pathways that govern intestinal cell function and molecular underpinnings are likewise conserved in *Drosophila* [9], including JAK-STAT, Wg/Wnt, Hippo, and epidermal growth factor receptor (EGFR) pathways. These homologies render *Drosophila* an exemplary model for the study of mammalian intestinal inflammatory diseases. The etiology of IBD is also linked to the gut microbiota, and pathogenic agents and microbial alterations that induce intestinal pathology in humans can similarly trigger intestinal lesions in *Drosophila* [9].

The current pharmacotherapies for IBD predominantly comprise 5-aminosalicylic acid (5-ASA) [10], which is notorious for its substantial adverse effects [11]. A burgeoning corpus of scientific literature has highlighted the potential of natural products derived from an array of botanical sources, which exhibit superior immunomodulatory, anti-inflammatory, and antioxidant capabilities, thereby offering a distinct advantage of reduced side effect profiles in the treatment regimen [12]. It is also of significance to note that numerous natural products mediate their therapeutic effects through the modulation of key cellular signaling cascades, including the JAK-STAT, TLR4/NF-κB, and mitogen-activated protein kinase (MAPK) pathways.

In this review, we will assess the feasibility of utilizing *Drosophila* as a model organism for the study of IBD by juxtaposing the structural homologies between the fruit fly’s intestine and that of mammals, as well as the congruence in the alterations of intestinal inflammation response pathways. Additionally, by aggregating the IBD-affiliated signaling pathways and the effects of natural products on various models under these cellular signaling cascades, we aim to investigate the pathogenic mechanisms of IBD and explore novel therapeutic avenues.

## 2. Intestinal Architecture of *Drosophila melanogaster* (Analogous to Mammals)

The intestinal structure, intestinal pathophysiology, and signaling pathways controlling physiological activities in fruit flies are largely similar and conserved compared to mammals [9]. The fruit fly gut is a highly compartmentalized tubular organ, composed of the foregut, midgut, and hindgut, which share structural similarities with the intestines of mammals (Figure 1) [13]. Food in fruit flies is temporarily stored in the crop within the foregut, analogous to the human stomach, where digestion and absorption occur in the midgut, similar to the function of the human small intestine, and water and electrolytes are exchanged in the hindgut, with excretion ultimately occurring at the anus [9].

The intestinal epithelium is the first line of defense against foreign invading microorganisms, with multiple defense mechanisms for detecting and eliminating invading pathogens, including the production of antimicrobial substances and signal transduction pathways [8]. The midgut is a significant immunological site in fruit flies, similar to the small intestine in mammals. Fruit flies and mammals have similarities in the composition and structure of intestinal cells; the fruit fly gut is lined by a single layer of intestinal epithelial cells, primarily composed of four cell types, including intestinal stem cells (ISCs), absorptive enterocytes (ECs), secretory enteroendocrine cells (EEs), and enteroblasts (EBs).

Mammals possess specialized goblet cells that secrete mucus and Paneth cells that produce antimicrobial peptides (AMPs), functions that are analogously fulfilled in the midgut of *Drosophila*, where both mucus and AMPs are generated [4]. The intestinal epithelial cells of *Drosophila* are safeguarded by a chitin-rich membrane and a nutrient-enriched surrounding matrix, which effectively segregate the microorganisms from the midgut epithelium, thereby preventing invasion by potentially pathogenic microorganisms, a role that is congruent with the function of mucus in mammals. Both *Drosophila* and mammals exhibit the presence of ISCs within their intestinal epithelium. These ISCs undergo mitotic division and asymmetric differentiation to give rise to EEs and ECs. However, in mammals, the differentiation of EEs and ECs from ISCs occurs within the crypt base, a structural element that is absent in *Drosophila* (Figure 2).

Alterations in the gut microbiota caused by environmental or dietary factors have been established as one of the mechanisms involved in the pathogenesis of IBD [14]. The normal intestinal microbiota has a positive impact on immune responses and plays an indispensable role in preventing inflammatory diseases, among which short-chain fatty acids (SCFAs) [15] are particularly important. These are produced by the fermentation of dietary fiber by the gut microbiota. SCFAs play a role in intestinal epithelial integrity and mucosal immune responses by inducing the secretion of pro-inflammatory factors (including IL-8, IL-1β, and TNFα) by intestinal epithelial cells [14]. At the same time, SCFAs provide nutrition and energy to the host, interact with the gut microbiota and host cells, and by increasing SCFA production, can lower the pH of the gut environment [16], thereby resisting the growth of potential pathogenic microorganisms. These characteristics work together to maintain balance in the gut. When SCFA levels are low, it may lead to dysbiosis of the gut microbiota, increased inflammatory responses, and subsequently, intestinal inflammatory diseases, including IBD. The gut microbiota plays an important role in maintaining gut homeostasis, providing nutrition to the host and playing a significant role in regulating local and systemic metabolism. There is a delicate balance in the gut between beneficial bacteria that secrete vitamins and harmful bacteria that secrete toxic substances, among which beneficial bacteria help maintain a healthy microbiota.

Probiotics are defined as “live microorganisms that, when administered in adequate amounts, confer a health benefit on the host” [17]. An increasing body of research indicates that probiotics can prevent and maintain intestinal health, playing a significant role in preserving the balance of microbial communities within the intestinal mucosa and host defense mechanisms. Their mechanisms of action primarily involve [15] the production of bactericidal proteins, a reduction in pro-inflammatory cytokine secretion, the generation of short-chain fatty acids (SCFAs) to stimulate the production of signaling proteins, and the enhancement of mucus production to improve the integrity of the intestinal barrier (Figure 3). The gut microbiota of *Drosophila* comprises 31 species [13], including *Lactobacillus* and *Acetobacter* genera and common probiotics in mammals such as *Bifidobacterium* and *Lactobacillus* genera are also present in fruit flies. Probiotics inhibit pathogen growth, modulate immune responses, and mediate immune reactions by releasing certain antimicrobial agents, such as lactic acid, acetic acid, and hydrogen peroxide. Like mammals, the gut microbiota in fruit flies affects metabolism and immune responses, for example, by influencing the renewal of midgut stem cells, producing antimicrobial peptides (AMPs) to combat invading microorganisms, and inducing immune responses.

## 3. *Drosophila* in IBD Modeling: Pros and Cons

*Drosophila* serves as a significant biological model for studying IBD, with its own unique advantages (Table 1). However, compared to other model organisms, fruit flies also have certain limitations (Table 2). It is important to note that no single model organism is the best for all purposes; each has its unparalleled strengths when addressing specific diseases and conditions.

## 4. The Impact of Microorganisms on the Gut Defense Pathways of *Drosophila*

The defense mechanisms against microbial invasion in the gut of *Drosophila* and mammals exhibit parallels. The intestinal epithelium of both organisms employs conserved innate immune pathways, such as the immune deficiency (Imd) and signal transducer and activator of transcription (STAT) pathways, to mount a resistance against bacterial encroachment. Additionally, the expression of AMPs and the production of ROS are instrumental in combating microbial invasion. The integrity of the intestinal barrier is a crucial factor in the defense against pathogenic microbes, and thus, the self-renewal and proliferation of ISCs are of significant interest. However, persistent inflammatory responses, continuous activation of these pathways, and excessive proliferation of ISCs may result in the infiltration of inflammatory cells into the intestinal mucosa and lead to uncontrolled immune response pathways, causing tissue damage within the gut epithelium.

### 4.1. Reactive Oxygen Species (ROS)

ROS, which encompass a spectrum of oxygen-derived molecules, are commonly produced through redox reactions or electronic excitation [22]

Within the physiological context of both *Drosophila* and mammals, ROS that arise from oxidative stress serve as pivotal mediators in antimicrobial defense, orchestrate tissue reparative processes, and modulate signal transduction pathways. A substantial body of research has demonstrated that the propagation of oxidative stress signaling is implicated in the pathogenesis of IBD [23]. During chronic intestinal inflammation, there is a pronounced infiltration of leukocytes into the mucosal cells, culminating in the excessive generation of ROS and the consequent induction of oxidative stress. This oxidative stress can compromise the cytoskeletal integrity; enhance intestinal permeability, further dismantling the intestinal barrier; and potentially alter the gut’s responsiveness to both symbiotic and pathogenic microbiota. Collectively, these findings underscore the significance of ROS, generated under conditions of oxidative stress, as a contributing factor in the etiology of IBD.

*Drosophila* and mammals utilize the production of ROS as a defense mechanism against microbial incursions. ROS are predominantly generated through the induction of two conserved flavoenzymes: Dual oxidase (Duox) and Nicotinamide adenine dinucleotide phosphate-oxidase (Nox). Notably, while Nox enzymes primarily produce hydrogen peroxide (H_2_O_2_), Duox enzymes possess the unique capability to generate hypochlorous acid in addition to H_2_O_2_ [24]. In mammals, an ortholog of *Drosophila* Duox is present, designated as Duox2. Duox has been implicated in the modulation of the interplay between the intestinal microbiota and the mucosal interface. Empirical evidence suggests that the activity of Duox is influenced by uracil levels; a decrease in intracellular calcium (Ca^2+^) concentration associated with low uracil levels results in diminished Duox activity [13]. Consequently, *Drosophila* with impaired Duox activity exhibit heightened susceptibility to intestinal pathogens. Prolonged inflammatory responses, along with the persistent activation of immune pathways and excessive proliferation of ISCs, can lead to the infiltration of inflammatory cells into the intestinal mucosa and result in unregulated immune response pathways. These processes may cause damage to the intestinal tissue, highlighting the role of ROS, particularly those generated under conditions of oxidative stress, as a contributing factor in the pathogenesis of IBD.

ROS are integral regulators of various physiological parameters [25], and their dysregulation is often implicated in human diseases, including inflammatory responses. Elevated concentrations of ROS can overwhelm tissue antioxidant defenses [26], leading to structural and functional damage to the intestinal mucosal barrier, exacerbating mucosal inflammation, activating pro-inflammatory signaling pathways, and damaging critical cellular components such as DNA, lipids, and proteins [27], thereby promoting the development of IBD. Notably, studies have indicated that the upregulation or downregulation of ROS levels does not significantly affect ISCs.

Numerous studies have demonstrated that ROS are pivotal modulators of IBD-associated signaling pathways (Figure 4). ROS can facilitate the activation of mitogen-activated protein kinase kinase kinase (MAPKKK), specifically apoptosis signal-regulating kinase 1 (Ask1) [28], which acts as an upstream sensor for the c-Jun N-terminal kinase (JNK) and p38 pathways. Elevated ROS levels within cells often correspond with heightened activity of the JNK and p38 pathways, and activated Ask1 further promotes the generation of ROS. Under conditions of oxidative stress, ROS promote the release of the transcription factor nuclear factor erythroid 2-related factor 2 (Nrf2) from Kelch-like ECH-associated protein 1 (Keap1) and its translocation to the nucleus, where it activates the expression of antioxidant genes (AGE) to modulate ROS levels [27]. NF-κB, a critical regulator in immune responses, is involved in the transcriptional control of immune-responsive genes. The inhibitor of κB (IκB) functions as a repressor of NF-κB activity [29], and ROS-mediated IκB degradation leads to the activation of the NF-κB signaling pathway. Once activated, NF-κB translocates to the nucleus, augmenting the transcription of target genes, which predominantly exert pro-inflammatory effects and are implicated in the pathogenesis of IBD [30].

### 4.2. Antimicrobial Peptides (AMPs)

AMPs exert antimicrobial activity and are integral to the maintenance of intestinal microbiota tolerance and the prevention of intestinal infections [31]. There is an association between IBD and a reduction in AMPs; these peptides defend against bacterial infections by disrupting bacterial membranes and modulate the intestinal microbiota composition. *Drosophila* and mammals elicit the production of AMPs in response to invading microbes through two principal pathways: the Toll pathway and the Imd pathway [32]. The Imd pathway initiates NF-κB signaling cascade, which operates in parallel to the NF-κB signaling cascade triggered by the Toll pathway, inducing the expression of a spectrum of overlapping yet distinct effector proteins, inclusive of AMPs. It is noteworthy that AMP production in *Drosophila* is compartmentalized [13], with regulation of AMPs hinging on NF-κB. While the Toll pathway serves as a pivotal activator of NF-κB, it does not exert this function in the midgut. The immune response within the midgut is predominantly modulated by the Imd pathway [33], which becomes activated upon the intracellular receptor PGRP-LR’s detection of bacterial diaminopimelic acid (DAP)-type peptidoglycan, thereby instigating the nuclear translocation of the NF-κB transcription factor Relish and the subsequent induction of distinct AMPs across various intestinal regions.

During intestinal infections, a diverse array of AMPs is induced through Imd signaling transduction to sustain intestinal homeostasis. Inflammatory damage to the intestinal epithelium activates immune responses, resulting in the upregulation of AMP expression. Exposure of *Drosophila* to dextran sulfate sodium (DSS) stimulation leads to a significant increase in the expression of AMPs and inflammatory cytokines [6]. Additionally, the expression of AMPs is closely linked to the composition of the intestinal microbiota; a decrease in AMP expression within the midgut can precipitate alterations in the intestinal microbiota, consequently contributing to intestinal pathologies.

### 4.3. Intestinal Regeneration (ISC Renewal)

The restoration and regeneration of intestinal tissue are crucial for the maintenance of intestinal homeostasis, with the self-renewal and reparative capabilities of the intestinal mucosa being highly contingent upon the proliferation and differentiation of ISCs [34]. IBD often results in significant impairment of intestinal barrier function; the integrity of this barrier is dependent on ISCs for equilibrium, and IBD can intensify oxidative stress within the gut. This increase in oxidative stress can diminish the antioxidant capacity of the endogenous antioxidant system, leading to oxidative stress-induced damage to biological macromolecules [7]. Consequently, the resolution of oxidative stress is pivotal in the context of inflammatory bowel diseases. Therefore, the autorenewal, proliferation, and differentiation of ISCs play a critical role in elucidating the pathogenic mechanisms of IBD. The turnover of ISCs is primarily influenced by various factors and associated signaling pathways. We will now expound on these two aspects.

#### 4.3.1. Various Factors

Numerous studies have indicated that chemical agents such as DSS and sodium dodecyl sulfate (SDS) can induce damage to the midgut [4], and extensive ROS can induce oxidative stress, leading to excessive proliferation of ISCs and disrupting the balance of intestinal homeostasis, which often exacerbates damage to the intestinal epithelium. Pathogenic and commensal microbes can promote the production of the cytokine unpaired 3 (Upd3) by the intestinal epithelium, which is analogous to the mammalian interleukin-6 (IL-6) homolog, and Upd3 can alter the niche function of ISCs as well as promote the proliferation and differentiation of ISCs [13]. Studies have also shown that in murine models, the Epstein–Barr virus (EBV) tends to elevate the levels of the pro-inflammatory factor IL-17A, suppress Toll-like receptors, and exacerbate colitis in mice [35]. Similarly, inflammation is also intensified in *Drosophila* induced by DSS [9]. An increasing body of research suggests that enteroendocrine cells (EECs) in both *Drosophila* and mammals secrete inflammatory cytokines and peptide hormones in response to the microbiota within the gut [36], modulating the innate immune system. For instance, in response to microbial invasion, *Drosophila* activates the Imd pathway mediated by the expression of tachykinin-related peptide (Tk) in EECs of the foregut and midgut, and mammals activate the NF-κB pathway mediated by the expression of Toll-like receptors in EECs [37], which plays a significant role in the regulation of IBD.

#### 4.3.2. Associated Signaling Pathways

*Drosophila* and mammals share conserved cellular signaling pathways that regulate the proliferation and differentiation of ISCs, which are modulated by the JAK/STAT, EGFR, Wingless/Wnt, and Hippo signaling pathways [4]. These pathways are implicated in the control of ISC proliferation and differentiation, as well as the preservation of intestinal homeostasis. Under steady-state intestinal conditions, ISCs are regulated by the EGFR/Ras/MAPK signaling cascade to undergo slow-cycling division and asymmetric division, giving rise to ISCs and EBs [38]. However, upon intestinal damage or infection, the proliferation of ISCs is markedly enhanced by the EGFR/Ras/MAPK pathway (Figure 5) and the JAK/STAT signaling pathway [39].

The JAK/STAT signaling cascade functions as a conduit, transducing extracellular cytokine cues to transcriptional alterations within the nucleus, and is implicated in numerous cellular processes, including the proliferation, differentiation, and migration of immune cells [4]. It is a requisite pathway for the proliferation and differentiation of ISCs [38], exerting a pivotal role in intestinal regeneration and the maintenance of homeostasis. The JAK/STAT pathway is activated by pro-inflammatory ligands (Upd2, Upd3) secreted by compromised cells, with its mammalian counterpart involving IL-6 and STAT3. Activation of this pathway facilitates rapid proliferation and differentiation of ISCs to drive epithelial regeneration; however, dysregulation or hyperactivation of the JAK/STAT pathway can result in excessive proliferation of ISCs and aberrant differentiation of enterocytes, leading to the exacerbation of intestinal epithelial pathology [4].

The EGFR signaling cascade is a proliferative pathway that is indispensable for the regeneration of the intestinal epithelium in *Drosophila* and is subject to regulation by the Hippo, Hedgehog, and JNK pathways [6]. The EGF ligand vein is expressed in the muscular sheath surrounding the intestinal epithelium, thereby providing a constitutive signal to activate the ERK protein kinase within ISCs [40]. Under conditions of tissue damage or stress, ECs secrete Upd3, which acts on adjacent cells to induce the production of EGF ligands, consequently activating the EGFR/Ras/MAPK signaling pathway in ISCs and triggering the transcription of target genes. The EGFR pathway synergizes with the JAK/STAT pathway within stem cells to foster ISC proliferation [41]. Although ISC proliferation is induced by both pathways, the influence of EGF ligands is more efficacious in promoting ISC regeneration [42]. Notably, FOS, a transcription factor essential for proliferation and mediated by both JNK and EGFR, integrates both mitogenic and stress signals within ISC proliferation. It has been demonstrated that JNK and ERK modulate the activity of FOS through the phosphorylation of distinct residues, thereby controlling ISC proliferation in a combinatorial fashion [40]. Aberrant activation of the EGFR pathway can lead to a rapid and excessive proliferation of ISCs, ultimately resulting in intestinal hyperplasia in *Drosophila*.

The Wnt and JNK signaling pathways, along with the insulin-like growth factor 1 receptor (INSR), regulate the proliferation of ISCs in mammals. The JNK pathway activates Wnt signaling and can potentially lead to tumorigenesis [9]. In mammals, INSR and platelet-derived growth factor (PDGF, known as PVF in *Drosophila*) control ISC homeostasis, promoting ISC division and guiding cell populations to change in response to aging or oxidative stress.

## 5. Impact of IBD-Elicited Inflammatory Responses on Signaling Pathways in *Drosophila*

*Drosophila* and mammals exhibit congruence in their pathway alterations and intestinal cell changes. The related pathways can be categorized into host immunity (Imd), stress (JNK), homeostasis (JAK/STAT, EGFR), and development (Notch).

### 5.1. JAK/STAT Pathway Balances ISC Self-Renewal and Proliferation

As mentioned earlier, the JAK/STAT pathway plays an important role in regulating ISC proliferation. At the same time, the JAK/STAT pathway (Figure 6) can transduce extracellular cytokines into the nucleus of cells, playing a significant role in many cellular processes, including the growth and differentiation of immune cells. When cells are infected or damaged, the JAK/STAT pathway balances the self-renewal and proliferation of stem cells [4]. If the JAK/STAT pathway is abnormal or overactivated, it can lead to autoimmune diseases and inflammation, including IBD [26].

In the context of IBD, damaged or pathogen-invaded enterocytes (ECs) release pro-inflammatory ligands such as Upd3 [13]. Upd3 activates the Domeless (Dome) receptor, which in turn activates the JAK transcription factor Hopscotch (Hop), and Hop then binds to the STAT transcription factor Stat92E. Phosphorylated Stat92E translocates to the nucleus to enhance the transcription of target genes [4]. The homologous pathway in mammals involves the cytokine IL-6 and STAT3.

### 5.2. The Imd Pathway: A Crucial Microbial Sensing System

The activation of the NF-κB signaling cascade is a prevalent reaction to intestinal damage or infection. In *Drosophila*, this encompasses the Toll pathway and the Imd pathway, which induce the production of a variety of AMPs. The Toll pathway is active in the foregut and hindgut, whereas the midgut immune response is predominantly regulated by the Imd pathway [13]. However, the NF-κB pathway initiated by Imd is paralleled by the NF-κB pathway triggered by Toll, leading to the generation of overlapping yet distinct AMPs. The Imd pathway serves as a crucial microbial sensing system in *Drosophila*, analogous to the Toll-like receptor (TLR) pathway in mammals [43]. It is commonly activated by the peptidoglycan monomer of Gram-negative bacteria (TCT) and the diaminopimelic acid (DAP)–type peptidoglycan present in both Gram-negative bacteria and Gram-positive bacilli, recognized by peptidoglycan recognition protein-LC (PGRP-LC) and receptor peptidoglycan recognition protein-LE (PGRP-LE) [44]. Meanwhile, PGRP-LF, PGRP-LB, and PGRP-SC [33] diminish the capacity to activate the Imd signaling pathway [13], thus preventing chronic immune response activation that could result in inflammatory damage.

The Imd signaling pathway (Figure 7), in conjunction with Duox enzymes, is integral to the antimicrobial response within the intestinal barrier. This pathway is characterized by its ability to generate AMPs that serve as a frontline defense against invading pathogenic microorganisms. The effective execution of the Imd pathway necessitates the signal-dependent processing and nuclear translocation of the PGRP-LC receptor, as well as the NF-κB family transcription factor Relish, which is analogous to its mammalian counterpart. Within this context, Pickle acts as an alternative nuclear IκB protein, selectively repressing the transcription of target genes by inhibiting the formation of Relish homodimers. This intricate regulatory mechanism ensures a precise and controlled antimicrobial response [29].

Inflammation is a crucial immune response in the body. As mentioned earlier, AMPs play a role in maintaining intestinal homeostasis. Flies lacking the Imd pathway often face higher bacterial loads and a lack of functional AMP responses, which may lead to alterations in the gut microbiota [33]. Changes in the gut microbiota can potentially cause intestinal diseases, such as IBD. However, abnormal activation of the Imd pathway can produce excessive AMPs that further damage the injured intestine. Studies have shown that miR-9a, miR-981, and miR-277 can negatively regulate the Imd pathway [45]. Research has indicated that EBV in the intestines of IBD patients enhances the Imd pathway through EBV DNA [3]. Therefore, the Imd pathway holds an important position in the treatment of IBD.

### 5.3. MAPK Pathway Modulates Inflammatory Cytokine Expression

In the realm of inflammatory mediators, the modulation of the MAPK pathway exerts a significant role in the regulation of pro-inflammatory cytokine expression [7], as well as in the processes of intestinal epithelial cell regeneration. The MAPK family encompasses a group of signaling molecules that are integral to the transduction of signals within the cell, including extracellular signal-regulated kinase (ERK), JNK, and p38 mitogen-activated protein kinase (p38). The typical MAPK pathways are activated through a sequential cascade reaction, and the MAPK signaling network comprises four primary routes, namely ERK, JNK, p38/MAPK, and ERK5 (Figure 8) [7]. Notably, the JNK and p38 pathways are implicated in the regulation of inflammation, apoptosis, and cellular differentiation and growth [46]. Upon intestinal mucosal injury, the MAPK pathway is activated and engages in the modulation of the expression of inflammatory cytokines, including the upregulation of Upd3.

The JNK pathway is a highly conserved branch of the MAPK pathways, composed of various MAPKs [6]. The JNK pathway is activated by both extrinsic and intrinsic stimuli, as well as by cytokines, primarily in response to stress factors [47]. In mammals, it is part of a complex kinase cascade, whereas in *Drosophila*, it is relatively simpler. The FOS family of transcription factors plays a crucial role in the JNK pathway by integrating mitogenic and stress signals, and it is particularly important in the process of ISC proliferation mediated by the JNK pathway [40]. When intestinal epithelial cells are damaged, the EC activate the JNK-induced ETS domain transcription factor Est21c expression, leading to the production of growth factors (such as EGF) and inflammatory factors (such as IL-6). Concurrently, the JNK/Ets21c pathway can induce EC apoptosis. Furthermore, under EB virus stress conditions, JNK is activated, prompting EB cells to produce IL-6. When ISCs receive signals from EC and EB, including EGF and IL-6, they activate the Ras/MAPK and JAK/STAT pathways, thereby promoting ISC proliferation [48].

The Ras/MAPK signaling pathway constitutes a distinct branch within the broader MAPK family. Aberrant or overactive immune signaling cascades are implicated in the pathogenesis of chronic inflammatory conditions, including IBD. The Imd pathway, which is a critical component of the innate immune response, serves as a frontline defense mechanism against microbial invasion. Empirical evidence suggests that the activation of the Ras/MAPK pathway can suppress the activity of the Imd pathway [49]. The underlying mechanism is believed to involve the activation of Ras/MAPK by receptor tyrosine kinases (RTKs), which in turn induces the transcription of Imd pathway inhibitors such as Pirk, Rudra, and PIMS, thereby attenuating Imd pathway activity. In the context of IBD, modulating the Ras/MAPK pathway may offer a therapeutic strategy to curb the excessive activation of the Imd pathway, thus preventing the exacerbation of inflammation.

Increasing evidence suggests that inflammation is associated with the suppression of DLG2 expression, which in turn is linked to the activation of inflammasomes and the release of cytokines [50]. The modulation of DLG2 expression may represent a potential mechanism for the improvement or treatment of IBD.

## 6. Comparison of the Effects of Natural Products on Different Biological Models in IBD Pathways

Natural products, as derivatives of secondary metabolism, are endowed with a spectrum of bioactive constituents and are distinguished by their minimal adverse effects and a range of biological activities, notably anti-inflammatory and antioxidant capabilities [51]. Those exhibiting therapeutic potential against IBD are predominantly classified into polyphenols, and terpenoids, among other chemical compounds. These natural products have been demonstrated to mitigate IBD symptoms through modulation of the intestinal microbiota [52], reduction in inflammatory responses [6], regulation of immune reactions [29], and amelioration of oxidative stress [26]. The pharmacological activities of natural products are conserved across different organisms, as evidenced by their effects in both *Drosophila melanogaster* and mammals [4].

At present, 5-ASA derivatives constitute a class of medications utilized in the treatment of IBD; however, research indicates that the chronic administration of 5-ASA can engender substantial side effects [51]. This has directed attention toward natural products, which are perceived to possess a lower propensity for adverse effects and are thus regarded as promising candidates for the prevention and management of IBD. To dissect the specific bioactivities and mechanisms underlying the actions of natural products, a plethora of biological models are employed for their screening and for elucidating their mechanisms of action in the context of IBD. These models conventionally engage the examination of signaling cascades, including but not limited to the TLR4/NF-κB, Nrf2/Keap1, MAPK, and JAK-STAT pathways. This comprehensive review will emphasize the comparative analysis of the effects of natural products within diverse biological models, such as murine, *Drosophila*, and human systems, as they pertain to IBD-associated pathways (Table 3).

A multitude of studies have demonstrated that signaling pathways play a significant role in the development of IBD, with natural products capable of modulating IBD symptoms through the regulation of signaling pathways, immune responses, and a reduction in inflammation. For instance, the NF-κB pathway is involved in the body’s inflammatory and immune responses, and its overactivation is often linked to the occurrence of human inflammatory diseases. As indicated in the aforementioned studies, various natural products, such as Luteolin and *Astragalus membranaceus* (AME) [65,82], ameliorate IBD symptoms by inhibiting the NF-κB pathway. The JAK/STAT pathway is crucial in regulating immune responses, inflammation, and intestinal regeneration, with natural products like olive, AME, total ginsenosides, and atractylodin [7,54,65,82], improving IBD induced by SDS in *Drosophila* by reducing the expression of p-JAK and p-STAT. Furthermore, AME and others inhibit the JAK/STAT pathway in mice induced by DSS, thereby modulating the gut microbiota by reducing the expression of pro-inflammatory factors. The MAPK pathway, a common inflammatory signaling pathway, is involved in maintaining the homeostasis of the intestinal epithelium. Studies have shown that the expression of proteins in the MAPK pathway is significantly increased in IBD, and natural products such as quercetin and olive [70,83] inhibit the MAPK pathway in mice induced by DSS, thereby suppressing the expression of pro-inflammatory factors such as tumor necrosis factor α (TNF-α), IL-1β, and IL-6.

From the above, it is evident that the gut microbiota occupies a significant position in the onset and progression of IBD. Natural products can modulate the equilibrium of gut microbiota by regulating signaling pathways and promoting the proliferation of probiotics. Dietary polyphenols can inhibit pathogenic microbes and maintain intestinal homeostasis by stimulating the growth of probiotics, such as *Lactobacillus* and *Bifidobacterium* [84], which in turn secrete short-chain fatty acids (SCFAs) [85]. In murine experiments, AME was found to ameliorate the symptoms of colitis by reversing the perturbation of gut microbiota induced by DSS. Caffeic acid also mitigates inflammation by restoring fecal microbiota abundance in DSS-induced colitic mice and suppressing the elevation of the *Firmicute* to *Bacteroidetes* ratio [86].

An increasing body of research indicates that natural products possess a variety of biological activities, including notably potent antioxidant and anti-inflammatory properties (Table 4). When intestinal cells are damaged, the activation of pro-inflammatory factors and signaling pathways stimulates the proliferation of ISCs. Excessive ISC proliferation can often lead to the exacerbation of inflammation. *Acanthopanax senticosus* polysaccharides (ASPS) [4] can improve intestinal homeostasis in *Drosophila* under DSS stimulation, reducing the overproliferation and differentiation of ISCs by inhibiting the EGFR and JNK pathways; Safranal, UA, CA, SIL, and AME can also inhibit ISCs’ disorder [4,54,79,86,87], preventing intestinal damage by suppressing the accumulation of ROS in the gut, among which UA extract can repair the abnormal morphology of ISC/EB.

In some experiments, natural products have been found to potentially benefit the lifespan of fruit flies. Polysaccharides from *Premna microphylla* Turcz (PPMT) [4] possess anti-inflammatory capabilities, and the application of PPMT significantly extended the lifespan of fruit flies induced with SDS. It is suggested that PPMT might enhance the expression of immune-related AMP pathways, mTOR pathways, and Imd pathways. In separate experiments with SDS-induced fruit flies, *Flos Puerariae* extract (FPE), SIL, and AME showed no toxic effects on the gut [34,54]. After administration of these substances, there was a reduction in ROS accumulation in the gut and inflammatory stress responses, with a notable extension of the lifespan in both SDS- and DSS-induced *Drosophila* [6,54].

## 7. Discussion

With the confines of this comprehensive review, we have systematically delineated the exploration of pivotal signaling mechanisms in IBD utilizing the *Drosophila* model. Our findings illustrate that signaling pathways are capable of modulating the initiation and progression of inflammation by impacting the intestinal defense mechanisms against microbial incursions, regulating intestinal homeostasis, and facilitating the production of bactericidal substances in *Drosophila*. Furthermore, we have demonstrated that natural products, including flavonoids, exhibit antioxidant and anti-inflammatory characteristics [82]. These compounds can modulate the gut microbiota, mitigate inflammatory responses, and regulate immune reactions by targeting cytokines, inflammatory mediators, and associated signaling cascades, thereby conferring protective and therapeutic benefits against IBD. Collectively, our data substantiate the utility of *Drosophila* as a significant model organism for the investigation of intestinal IBD pathways.

*Drosophila* and mammals share a complex set of intestinal immune mechanisms that effectively counteract the invasion of both symbiotic and pathogenic microorganisms [20]. This mechanism involves conserved innate immune pathways, such as the Imd and STAT pathways, which induce the expression of AMPs and the production of ROS [29,40], thereby exerting antimicrobial effects. When the intestine experiences oxidative stress, ROS are generated that possess antimicrobial properties, guide tissue repair, and participate in signal transduction. Elevated levels of ROS often promote the activation of the JNK, p38, Nrf2/Keap1, and NF-κB pathways, thereby regulating the expression of target genes.

In this context, the Nrf2 transcription factor translocates to the nucleus, stimulating the expression of AGEs while concurrently downregulating ROS levels. Additionally, during microbial invasion of the intestine, it is typically the DAP found in TCT, Gram-negative, and Gram-positive bacteria that activate the Imd pathway (analogous to Toll-like receptors in mammals [29]), initiating the NF-κB signaling cascade to induce the production of AMPs. Aberrant activation of the Imd pathway can lead to excessive AMP production, which may compromise the integrity of the intestinal barrier and exacerbate the progression of IBD.

Our results further reveal that the maintenance of intestinal homeostasis and the renewal of intestinal epithelial cells are intricately linked to various signaling pathways. Specifically, the Egfr/Ras/MAPK and JAK/STAT pathways play crucial roles in regulating ISC renewal during intestinal damage, while the Egfr/Ras/MAPK, JAK/STAT, EGFR, Wg/Wnt, and Hippo pathways collectively contribute to the regulation of intestinal homeostasis [40,44]. 

How do natural products modulate signaling pathways to ameliorate and prevent IBD? From the above summary, it was found that the polyphenolic compounds lignans and curcuminoids reduce the expression level of inflammatory factors mainly by inhibiting inflammation and oxidative stress, and inhibiting NF-κB activation. Among them, turmeric was able to inhibit STAT3 activity and IL-1β production, maintain the composition and homeostasis of intestinal flora, and inhibit NF-κB pathway and JNK phosphorylation in mice induced by DSS. Terpenoids such as UA and Sesamol have potent anti-inflammatory activity and antimicrobial properties, preventing IBD development and progression mainly by inhibiting pro-inflammatory mediators, reducing inflammatory responses and oxidative damage. Plant substances such as ginseng, *Hoveniae seu fructus*, FPE [59], etc. also have a variety of biological activities, including anti-inflammatory, anti-toxicity, immunomodulation, etc. Ginseng is able to inhibit the JNK/ERK pathway to improve the intestinal flora and stimulate intestinal stem cells to differentiate and proliferate, in which FPF plays its role as an anti-inflammatory agent by inhibiting the NF-κB, MAPK, and JAK/STAT pathways. Among them, FPE and its antioxidant and anti-inflammatory activities regulates pro-inflammatory factors and oxidative stress by inhibiting NF-κB, MAPK and JAK/STAT pathways.

In summary, natural products modulate a variety of signaling pathways to exert their protective effects against IBD, underscoring their potential as therapeutic agents in the management of these chronic inflammatory conditions.

## 8. Conclusions

*Drosophila melanogaster* and mammals exhibit conservation in intestinal architecture, signaling pathways, and gut microbiota. Natural products have a nonnegligible role in extending the lifespan of fruit flies, which can be utilized as a model for screening natural products due to their low cost and rapid reproduction rate. However, due to the absence of an adaptive immune system in fruit flies, they require collaborative studies with other model organisms for research on adaptive immunity, acting as a complementary biological model.

## Figures and Tables

**Figure 1 ijms-25-12742-f001:**
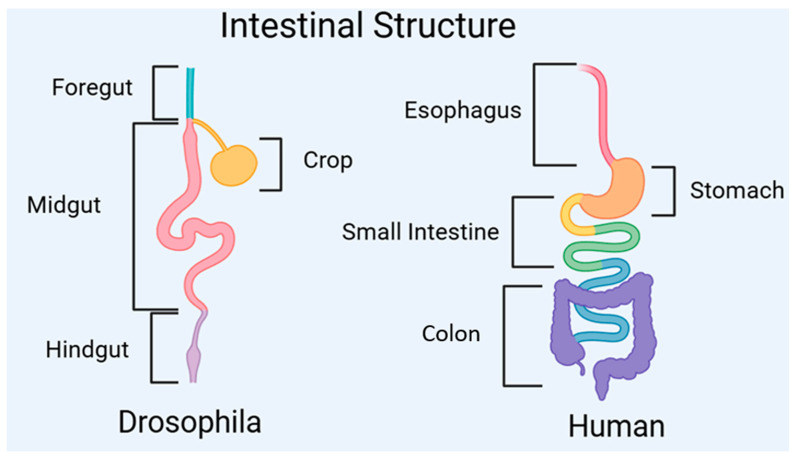
The gastrointestinal structures of *Drosophila* and mammals.

**Figure 2 ijms-25-12742-f002:**
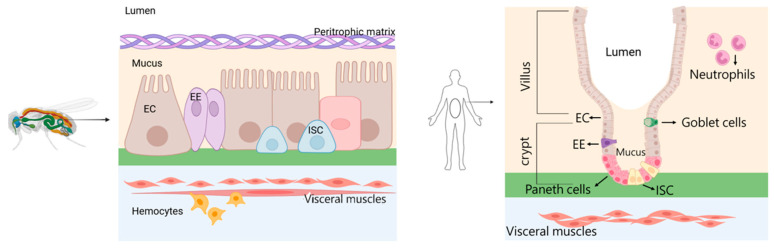
The constitution of intestinal cells in *Drosophila* and mammals.

**Figure 3 ijms-25-12742-f003:**
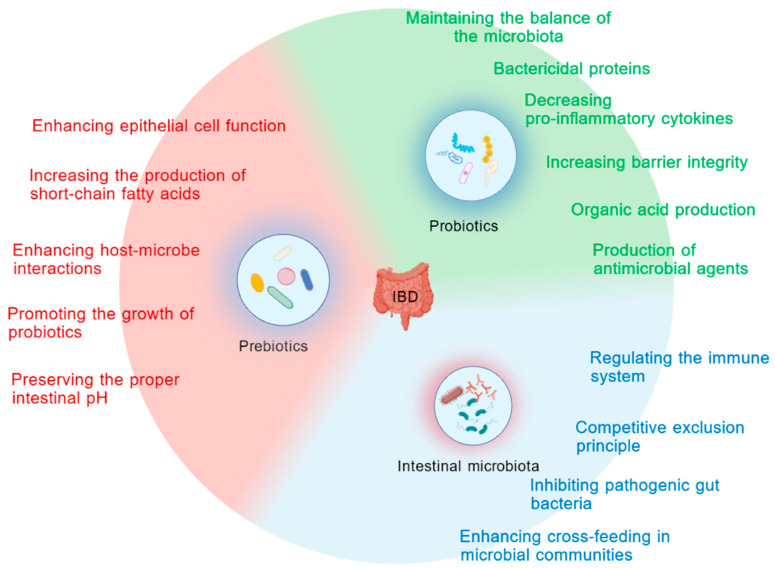
Mechanisms of action of probiotics and prebiotics in IBD.

**Figure 4 ijms-25-12742-f004:**
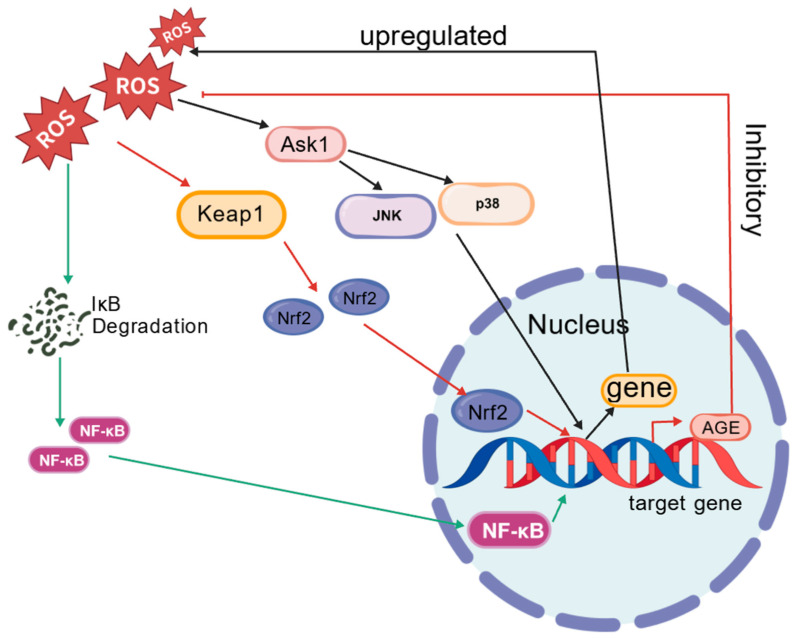
ROS and the regulation of key pathways (JNK/p38, Nrf2/Keap1, NF-κB).

**Figure 5 ijms-25-12742-f005:**
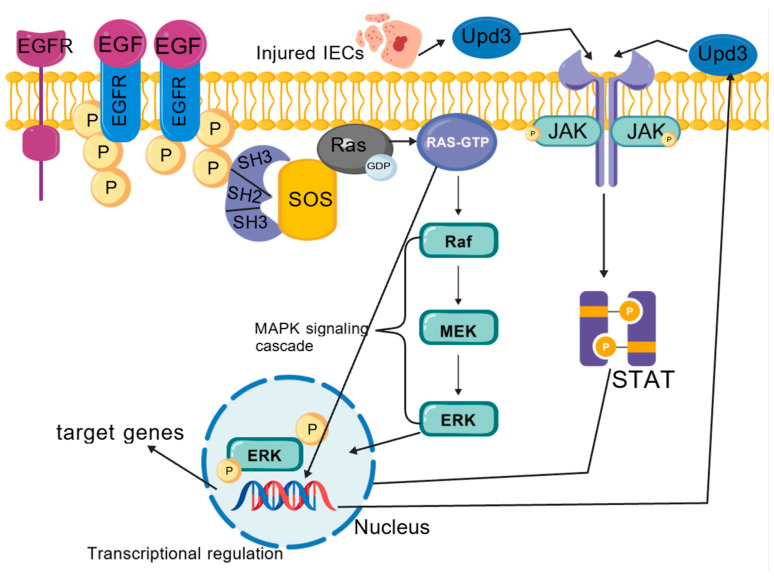
EGFR/Ras/MAPK and JAK/STAT pathways’ reg. on ISC.

**Figure 6 ijms-25-12742-f006:**
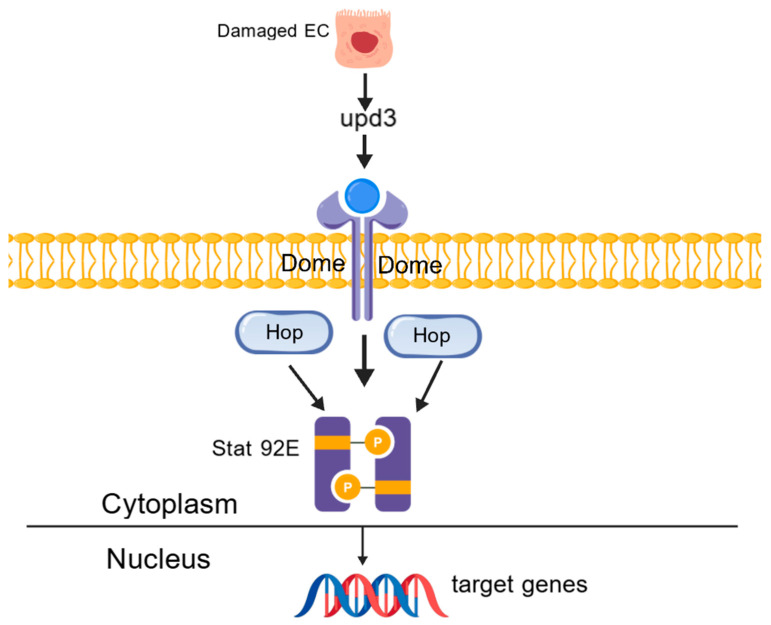
The mechanism of the JAK/STAT pathway.

**Figure 7 ijms-25-12742-f007:**
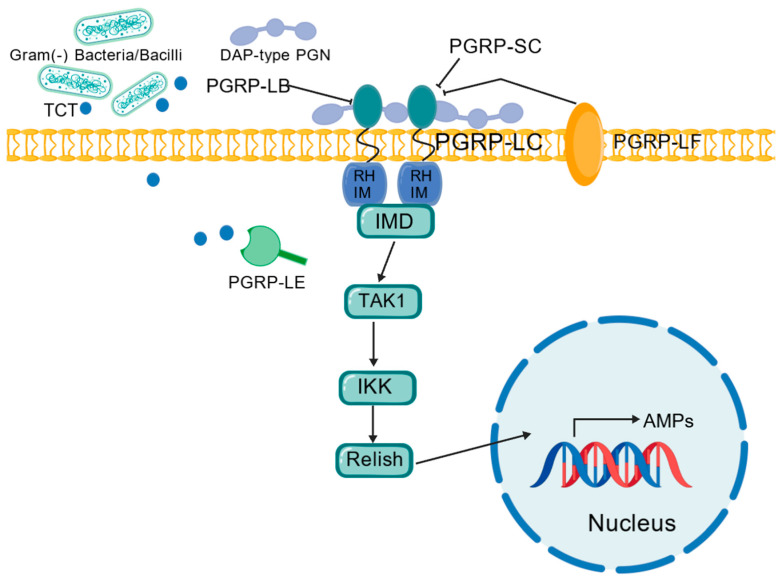
The mechanism of the Imd pathway.

**Figure 8 ijms-25-12742-f008:**
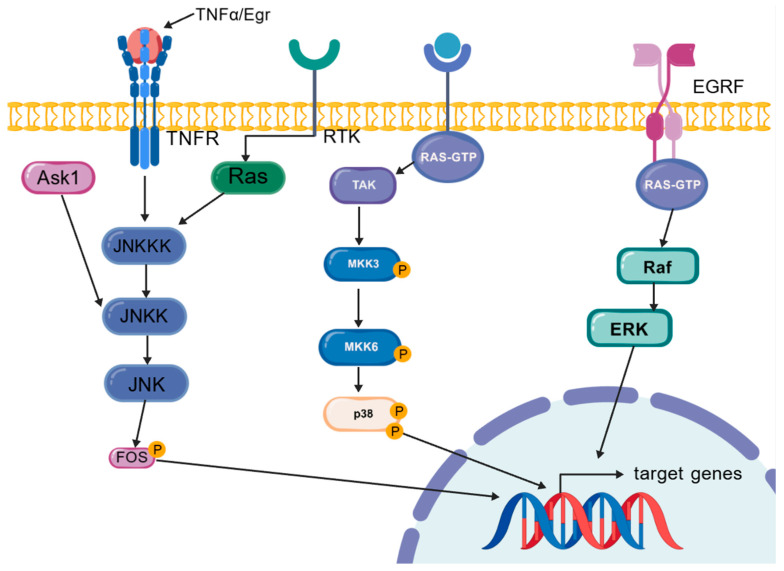
The mechanism of the MAPK pathway.

**Table 1 ijms-25-12742-t001:** Advantages of *Drosophila* over other models.

Gene	Genome sequenced [18], 75% human gene homology, simpler structure, and fewer paralogs in intestinal regeneration genes [9]
Intestinal structure	Similar in structure and function to the human intestine, a simple and highly conserved organ system [19]
Signal pathways	Conserved gut pathways help study stem cell actions and inflammation [20]
Gut microbiota	Simple microbiota aids in studying host-microbe interactions and disease origins in the gut. Pathogens causing human gut issues can similarly affect *Drosophila*
Immune response	Highly conserved with human innate immune responses, including the ways of detecting and responding to pathogens, as well as the genes involved in these processes
Drug screening	*Drosophila* help identify potential drugs with key features and overcome cell culture limitations in toxicity and pharmacological evaluation.
Cost	Affordable, easy, fast breeding, minimal ethics approval, short life cycle [18]

**Table 2 ijms-25-12742-t002:** Disadvantages of *Drosophila* as a model organism.

Distant phylogenetic relationship with humans	*Drosophila’s* distant relation to humans limits its ability to model specific biological processes and diseases
Lack of adaptive immune system	*Drosophila* lacks adaptive immune [21] factors like lymphocytes, restricting gut immunity studies, and misses key immune signaling branches.

**Table 3 ijms-25-12742-t003:** The impacts of natural products or medicinal formulas on intestine-related signal pathways in diverse biological models.

Categorization	Pathway	Natural Products	Humans	Mice	*Drosophila*
Plant material	Wnt/β-catenin	*Flos Puerariae*			Inhibitory Wnt [34]
	JNK/ERK	*G* *insenosides*	Inhibitory JNK/ERK [53]		Inhibitory JNK/ERK [7]
		*Astragalus membranaceus*			Inhibitory JNK/ERK [54]
		*Hoveniae seu fructus*			
		*Crocus sativus*	Inhibitory JNK/ERK [55]	Inhibitory JNK/ERK [55]	
	TLR4/NF-κB	*Astragalus membranaceus*	Inhibitory NF-κB [56]	Inhibitory NF-κB [56]	
		*Ginsenosides*	Inhibitory NF-κB [57]		
		*Atractylodes macrocephala Koidz.*	Inhibitory Nrf2 [58]	Inhibitory Nrf2 [58]	Inhibitory Nrf2/Keap1
		*Flos Puerariae*		Inhibitory NF-κB [59]	
		*Hoveniae seu fructus*		Inhibitory NF-κB [60]	
		*Crocus sativus*	Inhibitory NF-κB [55]	Inhibitory NF-κB [55]	
		*Curcuma longa*		Inhibitory NF-κB [61]	
	Nrf2/Keap1	*Flos Puerariae*			Stimulatory Nrf2/Keap1 [34]
		*G* *insenosides*		Stimulatory Nrf2 [62]	
		*Atractylodes macrocephala Koidz.*	Inhibitory Nrf2 [58]		
	MAPK	*Ginsenosides*		Inhibitory MAPK [53]	
		*Atractylodes macrocephala Koidz.*	Inhibitory MAPK [63]	Inhibitory MAPK [63]	
		*Crocus sativus*		Inhibitory MAPK [55]	
		*Hoveniae seu fructus*		Inhibitory MAPK [60]	
	JAK-STAT	*Astragalus membranaceus*			Inhibitory JAK/STAT [54]
		*Flos Puerariae*			Inhibitory JAK/STAT [34]
		*Atractylodes macrocephala Koidz.*	Inhibitory JAK/STAT [58]	Inhibitory JAK/STAT [58]	
		*Hoveniae seu fructus*		Inhibitory JAK-STAT	
		*Curcuma longa*		Inhibitory STAT3 [64]	
Compound	Wnt/β-catenin	Luteolin		Inhibitory Wnt/β-catenin [65]	
	JNK/ERK	Silybin		Inhibitory JNK [6]	
		Oleuropein		Inhibitory JNK [66]	
		Ursolic acid	Inhibitory JNK/ERK [67]		
	EGFR	Silybin			Inhibitory EGFR [6]
	TLR4/NF-κB	Luteolin		Inhibitory HMGB1-TLR-NF--κB	
		*Hydrolysate of yellowtail sperm*	Inhibitory NF-κB [68]		
		17β - estradiol		Inhibitory NF-κB [69]	
		Quercetin	Inhibitory NF-κB [70]		
		Ursolic acid	Inhibitory NF-κB [67]	Inhibitory NF-κB [67]	
		Silybin	Inhibitory NF-κB		
		Stilbenoid			Inhibitory NF-κB [71]
		Oleuropein	Inhibitory NF-κB [72]		
	Nrf2/Keap1	accinium vitis-idaea L			Stimulatory Nrf2 [52]
		Luteolin		Stimulatory Nrf2 [73]	
		Caffeic acid		Stimulatory Nrf2 [74]	
		Sesamin		Stimulatory Nrf2 [75]	
	Hedgehog	Silybin		Inhibitory Hedgehog [76]	
	MAPK	Luteolin			
		Oleuropein		Inhibitory MAPK [77]	
		Quercetin		Inhibitory MAPK [70]	
	JAK-STAT	Oleuropein	Inhibitory STAT3 [78]		
		Silybin	Inhibitory STAT3 [79]		
Decoction/Remedy	Notch	Compound sophorae decoction		Inhibitory Notch [80]	
	TLR4/NF-κB	Tongxie Yaofang	Inhibitory TLR4 [81]	Inhibitory TLR4 [81]	
	JAK-STAT	Tongxie Yaofang	Inhibitory JAK-STAT [81]	Inhibitory JAK-STAT [81]	

**Table 4 ijms-25-12742-t004:** Dose experiments of natural products in model organisms.

Natural Products	Cell Type/Model	Anti-Inflammatory Mechanisms	Experimental Group	Control Group	Time	Citation
Bilberry anthocyanin extracts	*Drosophila*	Relieve DSS-induced intestinal damage in fruit flies, reduce ROS levels, and promote the expression of Nrf2	**Inflammation:** Filter paper soaked in 7% DSS solution with 5% sucrose; **BANCs + Inflammation:** Filter paper soaked in a solution containing both 7% DSS and 0.1 mg/mL BANCs with 5% sucrose	**Control:** Filter paper saturated with only 5% sucrose, without the addition of BANCs or DSS; **BANC control:** Filter paper soaked in 5% sucrose containing 0.1 mg/mL BANCs.	Replace the filter paper every 12 h	[52]
*Astragalus membranaceus*	C57BL/6 mice	Relieve symptoms of DSS-induced colitis in mice, inhibit the NF-κB pathway, and significantly reduce the mRNA expression of IL-6, IL-8, and TNF-α in a dose-dependent manner	**AME 2 mg/kg**: 2 mg/kg of Astragalin, dissolved in the same volume of DMSO; **AME 5 mg/kg**: 5 mg/kg of Astragalin, dissolved in the same volume of DMSO.	**Control:** Filtered water; **Vehicle:** DMSO	subsequent 7 days	[56]
*Astragalus membranaceus*	HCT-116 and HT-29 human colon epithelial cells	Inhibit the proliferation of colon cells, suppress the expression of IL-6 mRNA induced by TNFα, and dose-dependently inhibit the phosphorylation and degradation of IκBα	HCT-116 and HT-29 cells were treated with various concentrations of AME (0, 20, 40, 60, 80, and 100 μM) for 24 h	/	/	[56]
*Astragalus membranaceus* extract	*Drosophila*	Inhibit abnormal intestinal stem cell (ISC) proliferation induced by SDS, reduce the accumulation of reactive oxygen species (ROS) in the intestine, suppress the JAK-STAT and JNK signaling pathways, and mitigate the extent of oxidative stress	**Different concentrations of AME (5, 10, 50 mg/mL):** AME was added to the standard cornmeal molasses medium at different concentrations	Standard cornmeal molasses medium	/	[54]
Caffeic acid	C57BL/6 mice	Inhibit the activation of the NF-κB signaling pathway, suppress the secretion of IL-6, TNFα, and IFNγ, restore the microbial community abundance in DSS-induced colitis mice, and inhibit the increase in the *Firmicute* to *Bacteroidetes* ratio	**DSS:** Sterile water is administered for the first 7 days, followed by water containing 2.5% DSS for the next 8 days; **DSS + CaA:** Water containing 1 mM CaA is administered for 15 days, and from the 8th day onwards, 2.5% DSS is added to the water	autoclaved water	subsequent 15 days	[86]
Resveratrol	BALB/c mice	Promote the increase in anti-inflammatory factors, inhibit pro-inflammatory factors, resveratrol regulates Treg/Th17 in a dose-dependent manner, and suppresses inflammatory responses in a dose-dependent manner	**MD:** 5% DSS is given for the first 7 days, followed by distilled water for the next 7 days; **RLD:** 5% DSS is given for the first 7 days, followed by distilled water for the next 7 days, and from the 7th day onwards, 50 mg/kg resveratrol in a 0.5% ethanol solution is administered daily via gavage; **RHD:** 5% DSS is given for the first 7 days, followed by distilled water for the next 7 days, and from the 7th day onwards, 100 mg/kg resveratrol in a 0.5% ethanol solution is administered daily via gavage.	sterile distilled water	subsequent 14 days	[88]
Anthocyanins extract	C57BL/6 mice	Relieve TNBS-induced colonic inflammation in mice, and inhibit the expression of IL-12, TNF-α, and IFN-γ in a dose-dependent manner	Different doses of anthocyanin extract (10, 20, and 40 mg/kg) groups: Anthocyanin extract at different doses was dissolved in 100 μL of physiological saline	**Carrier:** Rectal injection of 100 μL of 50% PBS; **Control:** A single rectal injection of 0.5 mg TNBS	once daily	[89]
Naringenin	C57BL/6 mice	Relieve DSS-induced colon inflammation in mice, inhibit TLR4 signaling, suppress the NF-κB signaling pathway, inhibit the expression of pro-inflammatory mediators, and suppress pro-inflammatory factors	**NG treatment:** 50 mg/kg NG; **DSS model:** Daily oral administration of 100 μL of 0.5% (*w*/*v*) methylcellulose, and from day 4 to day 10, 4% (*w*/*v*) DSS was given in drinking water; **DSS + NG treatment:** Daily oral administration of 50 mg/kg NG starting 3 days before DSS treatment and continuing until the end of DSS treatment.	Vehicle: 100 μL of 0.5% (*w*/*v*) methylcellulose	subsequent 10 days	[90]
Naringenin	Wistar albino rats	Relieve ulcerative colitis induced by 4% acetic acid, higher doses of naringenin reduce the expression levels of TNF-α, IL-1β, and IL-6, dose-dependently decrease ulceration, necrosis, and inflammation in colon tissue, increase antioxidant markers, and inhibit NF-κB	**NG100:** 100 mg/kg NG daily; **AA:** Acetic acid treatment to induce ulcerative colitis; **NG25 + AA:** 25 mg/kg NG daily and acetic acid treatment; **NG50 + AA:** 50 mg/kg NG daily and acetic acid treatment; **NG100 + AA:** 100 mg/kg NG daily and acetic acid treatment; **MES + AA:** 300 mg/kg mesalazine (MES) daily and acetic acid treatment	No special treatment	subsequent 7 days	[91]
*Flos Puerariae* extract	*Drosophila*	Improve SDS-induced inflammatory damage, inhibit abnormal proliferation of ISCs, suppress the expression of ROS, and inhibit the JAK/STAT pathway and Wnt pathway	NG1, NG5, NG10 groups: 1 mg/kg, 5 mg/kg, and 10 mg/kg of FPE were added to the standard cornmeal molasses medium, respectively	Standard cornmeal molasses medium	Starting from the first larval stage, fruit flies got varying FPE levels till the experiment’s end; adults received 0.4 mm ML385 and 0.6% SDS	[34]
Silibinin	*Drosophila*	Relieve inflammatory damage induced by DSS or BLM, inhibit the over-proliferation of ISCs induced by DSS or BLM, reduce high levels of oxidative stress, decrease the expression of AMPs and inflammatory factors, and inhibit the JNK and EGFR signaling pathways	**Different concentrations of SIL (0.1, 1, 10 μM):** Cultured in standard fruit fly media with 0.1, 1, and 10 μM SIL, respectively; **DSS:** Cultured in standard fruit fly media with 7% DSS; **DSS + SIL:** Cultured in standard media with 7% DSS and different concentrations of SIL; **BLM:** Cultured in standard media with BLM (25 μg/mL); **BLM + SIL:** Cultured in standard media with BLM (25 μg/mL) and different concentrations of SIL	Standard *Drosophila* medium	Administer the corresponding drug once daily	[6]
Total *Ginsenosides*	*Drosophila*	Mitigate SDS-induced inflammatory damage, reduce the extent of oxidative stress, and inhibit the MAPK signaling pathway	**0.5% SDS:** Medium with 0.5% SDS; **0.5% SDS + 0.5% TGGR:** Medium with 0.5% SDS and 0.5% TGGR; **0.5% SDS + 1% TGGR:** Medium with 0.5% SDS and 1% TGGR; **0.5% SDS + 2% TGGR:** Medium with 0.5% SDS and 2% TGGR	Standard *Drosophila* medium	Fed normally for 7 days, then starved for 4 h. From day 8 to 9, given solution-soaked filter paper for 48 h	[7]
Ursolic acid	*Drosophila*	Ursolic acid mitigates the destruction of the intestinal barrier by SDS, reduces the excessive proliferation and differentiation of ISCs induced by SDS, decreases the levels of ROS, inhibits the phosphorylation of JNK, and suppresses the JAK/STAT pathway	**NF-SDS:** Fed with normal food for 7 days, then fed with 5% sucrose and 0.5% SDS. **UA-SDS:** Fed with food containing 100 μM UA for 7 days, then fed with 5% sucrose and 0.5% SDS	**NF-SUC:** Fed with normal food for 7 days, then fed with 5% sucrose	Filter paper and solution replaced daily, experiment repeated 3 times	[87]
Ursolic acid	mouse lymphocytes	Inhibition of CD4+ T cells, CD8+ T cells, and B cell proliferation, suppression of cytokine secretion by T cells and macrophages, and inhibition of MAPK and NF-κB activation	Mouse lymphocytes were treated with various concentrations of UA (ranging from 0.25 μM to 5 μM) for 4 h, followed by a 72-h culture period	Cells were treated with 0.1% DMSO and cultured in a medium without ursolic acid	/	[67]
Ursolic acid	C57BL/6 mice	Mitigate intestinal damage in mice with CCL4-induced liver fibrosis, inhibit TNF-α, improve the damaged intestine and suppress inflammation, and improve the abundance of the microbiota and microbial dysbiosis	**CCl_4_:** Gavage with CCL4 diluted in 20% olive oil (2 mL/kg); **UA:** Gavage with CCL_4_ for 4 weeks initially, then co-administered with UA (40 mg/kg/day) for another 4 weeks; **RhoAi:** Intravenous injection of adeno-associated virus (AAV) 1 week prior, followed by biweekly CCL_4_ gavage	Treated with olive oil (2 mL/kg) via gavage	Twice a week for 8 weeks	[92]
*Curcuma longa*	HCT116 CRC cells	Inhibit mTOR signaling transduction in HCT116 CRC cells	Groups with different CUR doses (0–100 μM)	Treated with 70% ethanol	CUR and vehicle treatments at 24, 48, and 72 h	[93]
*Curcuma longa*	mice	Mitigate TNBS-induced colitis, with the activation of nuclear factor-κB in the colonic mucosa being suppressed, a shift from pro-inflammatory Th1 to anti-inflammatory Th2 pattern, a reduction in the degree of lipid peroxidation, and a decrease in the activity of serine proteases	Administer different doses of CUR (from 25 mg/kg to 300 mg/kg) daily via oral gavage	Apply 30% PBS	Once daily for 10 days	
*Curcuma longa*	BALB/c mice	Protecting BALB/c mice but not SJL/J mice from TNBS-induced colitis.	**2% CUR diet:** Diet containing 2% CUR; **TNBS enema:** TNBS (2 mg/kg in 50% ethanol) administered via rectal enema to induce colitis; **2% CUR diet/TNBS enema:** Fed a diet containing 2% CUR while simultaneously receiving TNBS enema	**Control diet:** Fed a diet without curcumin; **Control enema:** Received PBS	Diets started 2 days before TNBS and continued for 7 days until the end	[94]
*Curcuma longa*	rats	Mitigate TNBS-induced colitis, inhibit NF-κB, block the degradation of IκB proteins, and suppress the expression of pro-inflammatory cytokine messenger RNA	**0.5% SASP:** 0.5% SASP is administered starting 3 days before TNBS treatment; **2.0% CUR-P:** 2.0% CUR is administered starting 3 days before TNBS treatment; **2.0% CUR-A:** 2.0% CUR is administered immediately after TNBS treatment.	**ETHA:** Received 50% ethanol treatment and were fed a regular diet; **TNBS:** Received TNBS treatment and were fed a regular diet	Sacrifice the subjects two weeks after TNBS administration	[95]
*Curcuma longa*	Peripheral blood mononuclear cells from healthy donors	Inhibit the signaling pathway induced by IL-12, reduce the production of phosphorylated STAT4, and regulate the balance of Th1 and Th2 cell responses	**CUR pretreatment:** Cells were preconditioned with 20 μg/mL CUR at 37 °C for 30 min without subsequent stimulation; **CUR + cytokine stimulation:** Cells were preconditioned with 20 μg/mL CUR for 30 min, followed by incubation with cytokines (e.g., 10 ng/mL IFN-β or 0.1 μg/mL IL-12) at 37 °C for an additional 30 min	**Control:** No treatment is applied; **Stimulus control:** Cells are stimulated with the respective cytokines (10 ng/mL IFN-β or 0.1 μg/mL IL-12) only, without prior curcumin preconditioning	/	[96]
Nanoparticle *Curcuma longa*	BALB/c mice	Mitigate DSS-induced colitis, inhibit the activation of NF-κB in colon epithelial cells, suppress the expression of pro-inflammatory mediators, and increase the abundance of butyrate-producing bacteria and butyrate levels in feces	**N-CUR:** Diet containing 0.2% (*w*/*w*) nano-particle CUR; **DSS:** Distilled water with 3% *w*/*v* DSS; **DSS + N-CUR:** Distilled water with 3% *w*/*v* DSS and diet containing 0.2% (*w*/*w*) nano-particle CUR	normal diet	Nanoparticle CUR started 7 days before DSS and lasted until the experiment’s end	[61]
*Curcuma longa*	ICR mice	Inhibit STAT3 phosphorylation in the colon of DSS-induced mice and suppress p53 expression	**CLD:** Oral dosage of 0.1 mmol/kg CUR per day; **CHD:** Oral dosage of 0.25 mmol/kg CUR per day; **DSS:** Administered 3% *w*/*v* DSS in drinking water one week after treatment with vehicle or CUR; **DSS + CUR:** Continued administration of CUR (doses can be 0.1 mmol/kg or 0.25 mmol/kg) alongside 3% *w*/*v* DSS in drinking water	Treated only with the vehicle (0.05% Carboxymethyl Cellulose)	subsequent 7 days	[97]
Quercetin	Rag1-deficient mice	Alleviate inflammatory responses in mice with T cell-dependent colitis, improve gut microbiota, inhibit macrophage secretion of pro-inflammatory factors, and promote the expression of Nrf2	**QCN treatment:** After adoptive transfer of CD4+CD25-CD62L+ T cells, orally administered 10 mg/kg QCN every 3 days	**PBS control:** Mice were orally administered PBS every 3 days after the adoptive transfer of CD4+CD25-CD62L+ T cells.	Every 3 days for 7 weeks	[98]
Quercetin	Rag1-deficient mice	Alleviate inflammatory responses in mice with T cell-dependent colitis, improve gut microbiota, inhibit macrophage secretion of pro-inflammatory factors, and promote the expression of Nrf2	**QCN treatment:** During the DSS treatment period, orally administered 10 mg/kg QCN every 3 days; **BMDM transfer:** On days 1 and 4 of DSS treatment, received intravenous injections of BMDMs (Bone Marrow-Derived Macrophages) treated with QCN or PBS	**Vehicle control:** Received the same volume of vehicle	Administered every 3 days during DSS treatment	[98]
Quercetin	Wistar rats	Mitigate DSS-induced colitis, inhibit NOS activity and iNOS expression, and suppress NF-κB activity	**QCN prevention:** QCN was administered orally at doses of 1 and 5 mg/kg/day while receiving 5% DSS	**Non-colitis:** No flavonoid treatment and distilled water, without DSS; **Colitis control:** Received 5% DSS only	QCN started with DSS, once daily for 8 days	[99]
Quercetin	Wistar rats	Mitigate DSS-induced colitis, inhibit NOS activity and iNOS expression, and suppress NF-κB activity	**QCN treatment:** QCN (dosage of 1 mg/kg/day) was administered starting when the concentration of DSS was reduced, concurrent with DSS treatment	**Non-colitis:** No flavonoid treatment, received distilled water, without DSS; **Colitis control:** Received 5% DSS for the first 5 days, followed by a reduction to 2% DSS for the next 10 days	Quercetin daily, starting when DSS concentration is reduced, until the end of the experiment	[99]
Quercetin	Wistar rats	25 mg/kg was ineffective throughout the experiment, while 50 mg/kg and 100 mg/kg alleviated TNBS-induced colitis, inhibited MPO activity, and increased GSH expression levels	**SASP:** SASP (25 mg/kg, oral) and TNBS; **QCN 25 mg/kg:** QCN (25 mg/kg, oral) and TNBS; **QCN 50 mg/kg:** QCN (50 mg/kg, oral) and TNBS; **QCN 100 mg/kg:** QCN (100 mg/kg, oral) and TNBS	**Naive:** Given only saline solution, without TNBS treatment; **Control:** Given saline solution and TNBS	Began QCT and SASP daily after TNBS, for 11 days	[100]
*Astragalus membranaceus*	C57BL/6J mice	Relieve DSS-induced colitis in mice, inhibit pro-inflammatory cytokine expression, suppress NF-κB activation, and increase gut microbiota diversity	**DSS + PBS:** 3% DSS and PBS; **DSS + DMSO:** 3% DSS and DMSO; **DSS + AME50, DSS + AME75, DSS + AME100:** 3% DSS and different doses of AG (dissolved in 20 μL DMSO + 180 μL PBS, doses of 50 mg/kg, 75 mg/kg, and 100 mg/kg respectively); **DSS + 5-ASA:** 3% DSS and 50 mg/kg of 5-ASA	Drinking plain water and fed AIN-93M diet	Administered 200 μL of different solutions daily with 3% DSS for 7 days to induce colitis	[101]

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
