# Peer review of "Drosophila: An Important Model for Exploring the Pathways of Inflammatory Bowel Disease (IBD) in the Intestinal Tract"

_ijms, 2024, doi:10.3390/ijms252312742_

Round 1
Reviewer 1 Report
Comments and Suggestions for Authors
The present manuscript entitled “Drosophila: An Important Model for Exploring the Pathways of Inflammatory Bowel Disease (IBD) in the Intestinal Tract” led by Zeng et al show Drosophila as a model for IBD. The manuscript is well written. However, the following should be incorporated in the manuscript.
· Add a separate section on the advantages and disadvantages of Drosophila as a model for IBD. If possible, provide a table.
· There were several IBD models are reported in the literature. Therefore, authors should briefly discuss, how the Drosophila model is better than other models?
· Fecal matter is used for biomarker investigation in IBD. Please discuss relevant information on IBD in Drosophila.
· Probiotics have been extensively investigated in IBD. Authors have discussed gut microbiota's role in IBD. However, the inclusion of information related to probiotics or prebiotics in the management of IBD in Drosophila would be more advantageous.
Reviewer 2 Report
Comments and Suggestions for Authors
The paper presents cutting-edge information on the use of the fruit fly (Drosophila melanogaster) as an in vivo animal model for studying inflammatory bowel disease (IBD). It offers a compelling comparison of the structure of the fly's digestive system to that of mammals, and the characteristics of the inflammatory responses and inflammatory cascades within this intestinal tract. This research has the potential to significantly advance our understanding of IBD.
The work's idea is very interesting. It systematises current information regarding using the fruit fly as a model for studying IBD, a topic of immense importance in the field. The main weakness of the manuscript is the chaos in the phytochemical description given in Table 1 and the Discussion section. After necessary corrections, I suggest submitting the manuscript for proofreading by an experienced Phytochemist.
The manuscript not only meets the standards for a review paper. The language quality is acceptable, and the numerous figures not only increase the substantive value but also enhance the ease of reception of the work. However, the manuscript lacks a numerous data important for repetition of the work and verification of its scientific soundness.
Major issues:
Please add in the manuscript text:
- cite Figures 1-7 at appropriate places in the manuscript;
- add a list of abbreviations;
- add citations of relevant literature in the Discussion section, as this section currently has no cited literature;
- add Conclusions section.
Minor issues:
L12, 14, 55, 58-59, 61, 64, 79, 86, etc.: the Latin genus and species names of the fruit fly (Drosophila melanogaster) should be written in italics. Please check the whole manuscript thoroughly.
L104, 247, 332: if an abbreviation is used for the first time in the text of the manuscript, it should be explained here. However, if this abbreviation is used again in the text of the manuscript, it does not need to be explained again; it should be used consistently throughout the manuscript. Please check the whole manuscript thoroughly.
L126: Latin names of bacterial strains (Lactobacillus, Acetobacter) should be written in italics. Please check the whole manuscript thoroughly;
L163-164: please use subscripts, e.g., H2O2;
L168: please use superscripts, e.g., Ca2+;
L284, 321: please eliminate or translate the Chinese character in the manuscript text;
L423, 471: flavonoids are polyphenols. Please rephrase this sentence.
L503-504: phytochemical chaos. Please group natural compounds as in Table 1 after corrections.
L514-515: atractylodes is not a group of phytochemicals, such as terpenoids, but at most a plant containing this phytochemical group of compounds. Please rephrase this sentence.
L521-522: ginsenosides are also terpenoids, specifically triterpene saponins. Please combine these two paragraphs (L514-526) and describe them as different groups of terpenoids, such as iridoids (monoterpenoids), diterpenes, and triterpenes.
Table 1:
In the second column of Table 1 containing the list of natural compounds, please sort all natural compounds into:
- pure compounds, e.g. luteolin, silybin, ursolic acid, etc.;
- plant substances, e.g. Astragalus membranaceus, Panax ginseng, Puerariae flos, Hoveniae seu fructus, etc. (Latin names of plants should be written in italics);
- plant products, e.g. olive oil.
What is Compound Sophora flower decoction?
In the third column, please pay attention to the spaces between words, e.g. JNK inhibitors [47], etc.
Reviewer 3 Report
Comments and Suggestions for Authors The paper was well written, especially after the modifications. They had adequate amount of signaling pathway details in the article. The only thing that I wanted to change was to specify that the review is about drosophila. The first version of the paper led the reader to believe that the authors are reviewing human signaling.Author Response
Thanks for checking it out.
Reviewer 4 Report
Comments and Suggestions for Authors
1. I suggest that the authors expand on the specific role of each signaling pathway in the pathogenesis of IBD within the Drosophila model.
2. I suggest further validation of the antioxidant potential of natural products discussed.
3. I suggest distinguishing between the activation levels of the Imd and Toll pathways in response to inflammatory triggers.
4. I suggest conducting a dose-response study on the natural products used, such as luteolin or quercetin.
5. The authors need to examine the effects of natural product treatments on Drosophila gut microbiota composition.
6. The authors need to investigate whether the natural products influence intestinal stem cell (ISC) proliferation.
7. I suggest examining the anti-inflammatory effects more specifically in Drosophila gut tissues. A minor ELISA or immunofluorescence for cytokines like IL-6 could demonstrate how natural products affect localized inflammation.
8. I suggest assessing the potential longevity benefits of antioxidant treatments. Conducting a lifespan study in treated Drosophila could provide initial evidence on whether these compounds extend healthspan under inflammatory conditions.
Round 2
Reviewer 1 Report
Comments and Suggestions for Authors
Authors have addressed all the issues satisfactorily
In the newly added probiotics image capitalize the i in intestinal microbiota
Author Response
感谢您指出这一点,我已经在新添加的益生菌作用机制图中制作了 I 大写
Reviewer 2 Report
Comments and Suggestions for Authors
The manuscript, now in its revised version, has been resubmitted to the International Journal of Molecular Sciences. While many changes have been incorporated, it's crucial that the present version aligns with the publication requirements. The primary area that needs attention is the phytochemical description in Table 3 (which should be Table 2). These necessary corrections are vital for the manuscript's publication.
Notes to be taken into consideration:
- please write the Latin name Drosophila melanogaster in italics everywhere in the manuscript;
- please correct the error in the numbering tables in the manuscript text. Where is Table 2 now in the manuscript? After adding Table 1 to the manuscript, the former Table 1 should become Table 2, not 3, etc. It is not the role of the Reviewer to draw the attention of the Authors to such editorial errors;
- Table 3: there is no Compound Sophora flower decoction. There is at most Sophora japonica flower decoction, which is not a compound but a plant extract, or more precisely, a decoction;
- Table 3: Latin names of plants and herbal substances, such as Puerariae flos, should be written in italics;
- the Authors have deliberately omitted the Reviewer's comment regarding phytochemical chaos in Table 3 (which should be Table 2). In the second column of Table 3 (which should be Table 2) containing the list of natural compounds, the Authors must sort all natural compounds into:
ü pure compounds, e.g., luteolin, silybin, ursolic acid, etc.;
ü plant substances, e.g., Astragalus membranaceus, Panax ginseng, Puerariae flos, Hoveniae seu fructus, etc. (Latin names of plants should be written in italics);
ü plant products, e.g., olive oil.
Author Response
Comment1:please write the Latin name Drosophila melanogaster in italics everywhere in the manuscript;
Respond1:Thanks for pointing this out, I have italicised all of the Drosophila melanogaster in the manuscript.
Comment 2:please correct the error in the numbering tables in the manuscript text. Where is Table 2 now in the manuscript? After adding Table 1 to the manuscript, the former Table 1 should become Table 2, not 3, etc. It is not the role of the Reviewer to draw the attention of the Authors to such editorial errors;
Respond 2:Thank you very much for pointing this out, I have corrected the error in the numbered tables of the manuscript, which has 4 tables. Table 1 is cited in L67, Table 2 is cited in L168, Table 3 is cited in L487, and Table 4 is cited in L524.
Comment 3:Table 3: there is no Compound Sophora flower decoction. There is at most Sophora japonica flower decoction, which is not a compound but a plant extract, or more precisely, a decoction;
Respond 3:I agree that there is no Compound sophorae decoction, but rather Compound sophorae decoction . I have amended this and categorised it as a decoction.
Comment 4:Table 3: Latin names of plants and herbal substances, such as Puerariae flos, should be written in italics;
Respond 4:Thank you very much for pointing that out, I have italicised Puerariae flos etc.
Comment 5:the Authors have deliberately omitted the Reviewer's comment regarding phytochemical chaos in Table 3 (which should be Table 2). In the second column of Table 3 (which should be Table 2) containing the list of natural compounds, the Authors must sort all natural compounds into:ü pure compounds, e.g., luteolin, silybin, ursolic acid, etc.;ü plant substances, e.g., Astragalus membranaceus, Panax ginseng, Puerariae flos, Hoveniae seu fructus, etc. (Latin names of plants should be written in italics);
Respond 5:Thank you very much for pointing this out, I have reworked Table 3 to categorise it into 3 categories, compounds, plant substances and decoction/remedies.
Reviewer 4 Report
Comments and Suggestions for Authors
No more comments
Author Response
谢谢你指出这一点。
Round 3
Reviewer 2 Report
Comments and Suggestions for Authors
The manuscript has been revised following the Reviewers' recommendations and is now suitable for publication in the International Journal of Molecular Sciences.